# Rock-Solubilizing Microbial Inoculums Have Enormous Potential as Ecological Remediation Agents to Promote Plant Growth

**Zhaohui Jia [1], Miaojing Meng [1], Chong Li [1], Bo Zhang [2], Lu Zhai [2], Xin Liu [1], Shilin Ma [1], Xuefei Cheng [1] and Jinchi Zhang [1,\*]**

[1] Co-Innovation Center for Sustainable Forestry in Southern China, Jiangsu Province Key Laboratory of Soil and Water Conservation and Ecological Restoration, Nanjing Forestry University, 159 Longpan Road, Nanjing 210037, China; zhjia2018@njfu.edu.cn (Z.J.); miaojingmeng@njfu.edu.cn (M.M.); cli5104@njfu.edu.cn (C.L.); liuxinswc@njfu.edu.cn (X.L.); mashilin@njfu.edu.cn (S.M.); CaroC1127@outlook.com (X.C.)

[2] Department of Natural Resource Ecology and Management, Oklahoma State University, Stillwater, OK 74078, USA; bozhang@okstate.edu (B.Z.); lu.zhai@okstate.edu (L.Z.)

\* Correspondence: zhang8811@njfu.edu.cn

**Abstract:** Anthropogenic overexploitation poses significant threats to the ecosystems that surround mining sites, which also have tremendous negative impacts on human health and society safety. The technological capacity of the ecological restoration of mine sites is imminent, however, it remains a challenge to sustain the green restorative effects of ecological reconstruction. As a promising and environmentally friendly method, the use of microbial technologies to improve existing ecological restoration strategies have shown to be effective. Nonetheless, research into the mechanisms and influences of rock-solubilizing microbial inoculums on plant growth is negligible and the lack of this knowledge inhibits the broader application of this technology. We compared the effects of rock-solubilizing microbial inoculums on two plant species. The results revealed that rock-solubilizing microbial inoculums significantly increased the number of nodules and the total nodule volume of *Robinia pseudoacacia* L. but not of *Lespedeza bicolor* Turcz. The reason of the opposite reactions is possibly because the growth of *R. pseudoacacia* was significantly correlated with nodule formation, whereas *L. bicolor*'s growth index was more closely related to soil characteristics and if soil nitrogen content was sufficient to support its growth. Further, we found that soil sucrase activity contributed the most to the height of *R. pseudoacacia*, and the total volume of root nodules contributed most to its ground diameter and leaf area. Differently, we found a high contribution of total soil carbon to seedling height and ground diameter of *L. bicolor*, and the soil phosphatase activity contributed the most to the *L. bicolor's* leaf area. Our work suggests that the addition of rock-solubilizing microbial inoculums can enhance the supply capacity of soil nutrients and the ability of plants to take up nutrients for the promotion of plant growth. Altogether, our study provides technical support for the practical application of rock-solubilizing microbes on bare rock in the future.

**Keywords:** rock-solubilizing microbial inoculums; ecological restoration of carbonate mining area; plant growth; root nodule

## 1. Introduction

Although open-pit mining makes significant contributions to economic development, it can also degrade ecosystems through soil erosion and the loss of productive land, which ultimately restricts social development and endangers the wellbeing of neighboring residents [1–4]. The ecosystems that surround mining sites face immense threats due to anthropogenic overexploitation. For instance, the carbonate rock area of China is 1.37 million km$^2$, which accounts for approximately 14.3% of its land area [5]. Ecological restoration, which reduces the hazards associated with the rocky desertification of mines [6–8] is critical, albeit challenging.

There are various approaches for the ecological restoration of carbonate mining areas. Soil spray-sowing is an ecological technology that uniformly sprays plant seeds and growth substrate materials on sloped surfaces under high pressure [9], which is one of the most widely employed strategies, particularly for the greening of rocky slopes [10]. However, issues related to the erosion resistance of bare rocky slopes when combined with spraying matrices occur subsequent to the application of this technology, which limits its applications for the ecological restoration of mining areas [11]. This is primarily manifested through difficulties in the sustained preservation and propagation of vegetation [12].

Alternatively, there has been a greater focus on the application of microorganisms for the remediation of mining sites [13,14]. Microbial flora can accelerate rock erosion processes; thus, improving the available soil nutrient content and enhancing slope stability by facilitating the fusion of rock surfaces with spraying matrices [15]. They can establish an interface between the rock mass and spraying substrate, which partially resolves the issue mentioned above. The application of microbes can further promote the permanent fixation of planting substrates on rock surfaces [16]. Therefore, efficient rock-solubilizing microbes are conducive toward the long-term maintenance of spray-sowing technologies. This can be beneficial for long-term ecological restoration effects [17] and more extensively employed for the restoration of ecosystems that surround mining sites [18].

Although the phytostabilization of mine tailings and mining sites is very promising and has been broadly employed worldwide [16,19], the effects of rock-solubilizing microbes on the long-term health and growth of plants remains unknown. In particular, since nitrogen is the main limiting nutrient for the long-term maintenance of slope residing plants [20], we endeavored to identify the effects of the addition of microbial inoculums on the nitrogen-fixing capacity of slope plants to provide a comprehensive understanding of the long-term effects of rock-solubilizing microbial inoculum on plant growth. We tested two hypotheses, including that the addition of rock-solubilizing microbes would (1) have a stimulating effect on plant growth with a focus on the optimal promotion of single strains and (2) increase the efficiency of nodule formation; thus, improving the capacity for nitrogen fixation. Our investigations may serve to guide further studies with the aim of enhancing the efficacy and benefits of soil spray-sowing technologies, while providing basic theoretical support for the ecological restoration of carbonate mining sites on a global scale.

## 2. Materials and Methods

### 2.1. Study Site

Pot experiments were conducted at the FYS-8 intelligent greenhouse at Nanjing Forestry University, China, where the relative humidity was 60%, and the maximum photosynthetically active radiation was 1800 mol/(m$^2 \cdot$s). The weighing method was adopted at 6:00 a.m. every day to replenish water and ensure that the soil moisture content of each pot attained 100% of the field water capacity (the water condition of each pot was consistent).

### 2.2. Seed Test Material, Microbial Strains, and Soil

The seeds of *Robinia pseudoacacia* L and *Lespedeza bicolor* Turcz were purchased from the Tianhe nursery garden company (Jiangsu, China). The germinated seeds were transferred to a seedling-raising disk and cultivated with nursery substrates. *Robinia pseudoacacia* and *Lespedeza bicolor* are both shallow-rooted plants with well-developed root systems. Because of their high stress resistance and strong soil-fixing ability, they are often employed in ecological slope restoration to improve slope stability and reduce soil loss. Therefore, these two plants were selected for this study to test the effects of rock-solubilizing microbes and growth substrates on plant growth.

*Bacillus thuringiensis* (NL-11, bacterial), *Streptomyces thermocarboxydus* (NL-1, actinomycetes), and *Gongronella butleri* (NL-15, fungal) were isolated from soil surrounding weathered dolostones [15,17]. These microorganisms were introduced into the liquid culture medium, fermented by shaking flasks for 24 h. Subsequently, the microbial seed fluid

was transferred to the fermentation tank. During the fermentation process, the microbes were extracted at predetermined intervals to determine their $OD_{600}$ value [13]. When the change curve attained its peak and began to decline, the microbes were transferred to a sterilized plastic bottle and stored in a refrigerator. In our previous research, we found that the addition of NL-11, NL-11 + NL-15, and NL-1 + NL-11 + NL-15 resulted in significant changes in the photosynthetic system and roots of *R. pseudoacacia* and *L. bicolor* [21]. Therefore, we selected these three inoculants and treatment methods to investigate the responses of plant growth and root nodulation.

Subsoil (20–50 cm) was collected from the Xiashu Forestry Station (32°7′47″ N, 119°13′15″ E). The soil type was yellow brown soil, and soil texture was clay loam. The soil characteristics included available K, 100.25 mg·kg$^{-1}$ and available P, 9.89 mg·kg$^{-1}$, at a pH of 7.25. Prior to the pot experiments, the soil was sifted through a 5-mm sieve and mixed with wood fiber, organic fertilizer, peat soil, and dolostones rock powder as nursery substrates (soil/wood fiber/organic fertilizer/peat soil/rock powder, 92:0.7:5:2:0.3) [21].

### 2.3. Pot Experiment Setup

Prior to the experiments, the pots (19.5 cm deep and 29.5 cm diameter) were washed with tap water. Each pot was filled with 5 kg of nursery substrates and 60 mL of microbe inoculums (the total amount of mixed microbes inoculum was 60 mL). Four microbial inoculums treatments: NL-11 (RPJ1), NL-11 + NL-15 (RPJ2), NL-1 + NL-11 + NL-15 (RPJ3), and sterilized microbial liquid culture medium (Control check, CK). Two tree species, *Robinia pseudoacacia* and *Lespedeza bicolor*, were used with each group. The experiment comprised eight treatment groups, each treatment group had five replicates, totaling 40 pots.

The seeds were cultured in a greenhouse from December 2018 to November 2019. We extracted root samples in November, which were placed with their original soil in insulated boxes on ice and quickly transferred to the laboratory to maintain freshness. The roots with nodules were rinsed clean in the laboratory, and the original soil was saved for determination of soil properties.

### 2.4. Plant Root Nodule Parameters

The root systems were scanned using a LA2400 Scanner (Expression 12000XL, EPSON, Long Beach, CA, USA). The scanned images had an 800 dpi resolution and were used for two-dimensional image analysis. The root and nodule traits, including the total root volume, nodule numbers, and nodule volume, were considered for analysis. Each root system was separated from the root nodules, which were placed in an oven at 40 °C and dried to a constant weight. The dried root nodules were placed on a scale and weighed to obtain the dry weight of the nodules.

### 2.5. Analysis of Soil Properties

The air-dried soil was ground to pass through a 1 and 0.15 mm sieve, respectively. The soil pH was measured with the soil and water at a ratio of 1:5 (*w:v*) using a pH meter and conductivity instrument, respectively [19]. The carbon (C) and nitrogen (N) contents of the soil were determined using an elemental analyzer (Vario EL III, Elementar).

The activities of the enzymes in soil related to the transformation of C, N, and P, i.e., soil catalase, sucrase, urease, and alkaline phosphatase were tested. The method employed for analysis was based on [22]. In brief, the sucrase activity was determined via 3, 5-dinitro salicylic acid colorimetry using sucrase as the substrate and expressed as micromole glucose per gram of dry sample. The urease activity was determined via indophenol colorimetry using urea as the substrate and expressed as micromole ammonium per gram of dry sample. The phosphatase activity was determined by disodium phenyl phosphate colorimetry and expressed as micromole phenol per gram of dry sample. The catalase activity was titrated over 20 min using 0.1 mol·L$^{-1}$ KMnO$_4$ and expressed as micromole KMnO$_4$ per gram of dry sample.

*2.6. Data Analysis*

Data analysis was done using Microsoft Excel 2010 and the SPSS package (version 21.0, IBM, Armonk, NY, USA), and the data were expressed as a mean ± standard deviation (SD). We analyzed data for normality and homogeneity and confirmed that the assumptions of normality and homogeneity were met, and the graphs were developed by Origin 8.0 (OriginLab Corporation, Northampton, MA, USA). The significance of different treatments on various indices was evaluated by ANOVA analysis with LSD's multiple comparisons, taking $p \leq 0.05$ as a significance level. Boosted regression tree (BRT) analysis was conducted using the R package "gbm" [23], and some of figures were generated using the R package "ggplot2" [24].

## 3. Results

### 3.1. Plant Growth Influenced by the Supply of Rock-Solubilizing Microbes

The RPJ2 and RPJ3 treatments significantly increased the height of *R. pseudoacacia* (Figure 1a, $p < 0.05$). The RPJ1 treatment augmented the height of *R. pseudoacacia* by 19% compared with the CK, but no significant difference was observed. However, the situation was different in promoting the height of *L. bicolor*. The RPJ1 treatment significantly increased the height of *L. bicolor* (Figure 1a, $p < 0.05$).

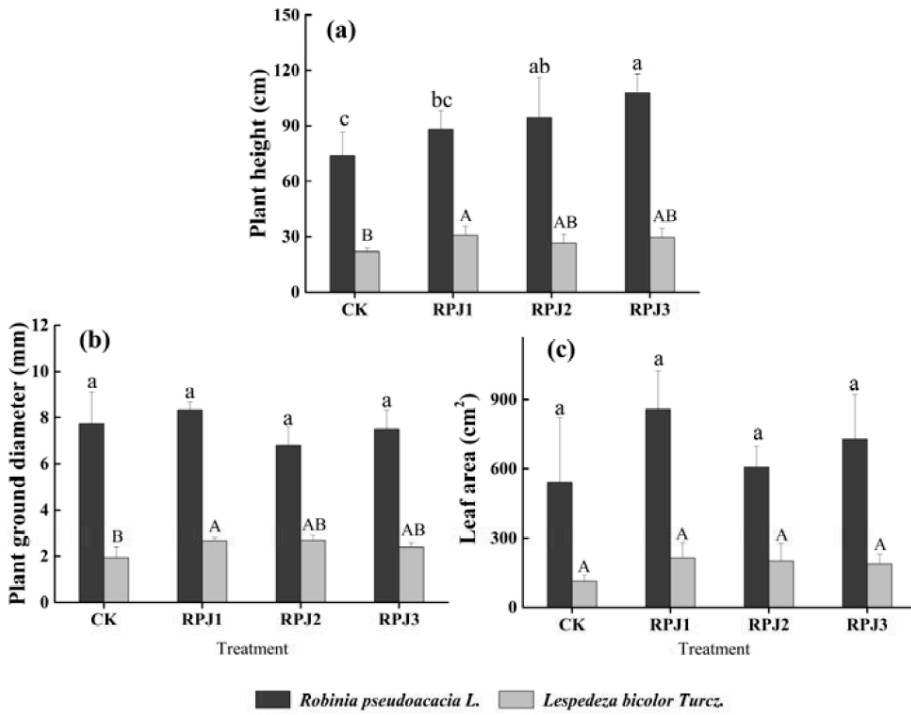

**Figure 1.** Changes in plant height (**a**), leaf area (**b**), and plant basal diameter (**c**) under the application of applied rock-solubilizing microbes. The data were presented as the mean ± SD (*n* = 3). One-way ANOVA (analysis of variance) was performed for each parameter. Different lowercase letters for *R. pseudoacacia* and capital letters for *L. bicolor* indicate a significant difference at $p < 0.05$ by Duncan's multi-range test.

The addition of rock-solubilizing microbes had no significant effect on the basal diameter and leaf area of *R. pseudoacacia* (Figure 1b,c, $p > 0.05$). However, in *L. bicolor* pots, the RPJ1 and RPJ2 treatment significantly enhanced the basal diameter of *L. bicolor* (Figure 1b, $p < 0.05$). The addition of rock-solubilizing microbes had no significant effect on the leaf area of *L. bicolor* (Figure 1c, $p > 0.05$).

### 3.2. Nodule Formation under the Addition of Rock-Solubilizing Microbes

As can be seen, the RPJ1 treatment significantly increased the root nodule number, root nodule volume, and root nodule contribution of *R. pseudoacacia* (Figure 2a,b,d, $p < 0.05$). However, the addition of rock-solubilizing microbes had no significant influence on dry weight root nodules of *R. pseudoacacia* (Figure 2c, $p > 0.05$).

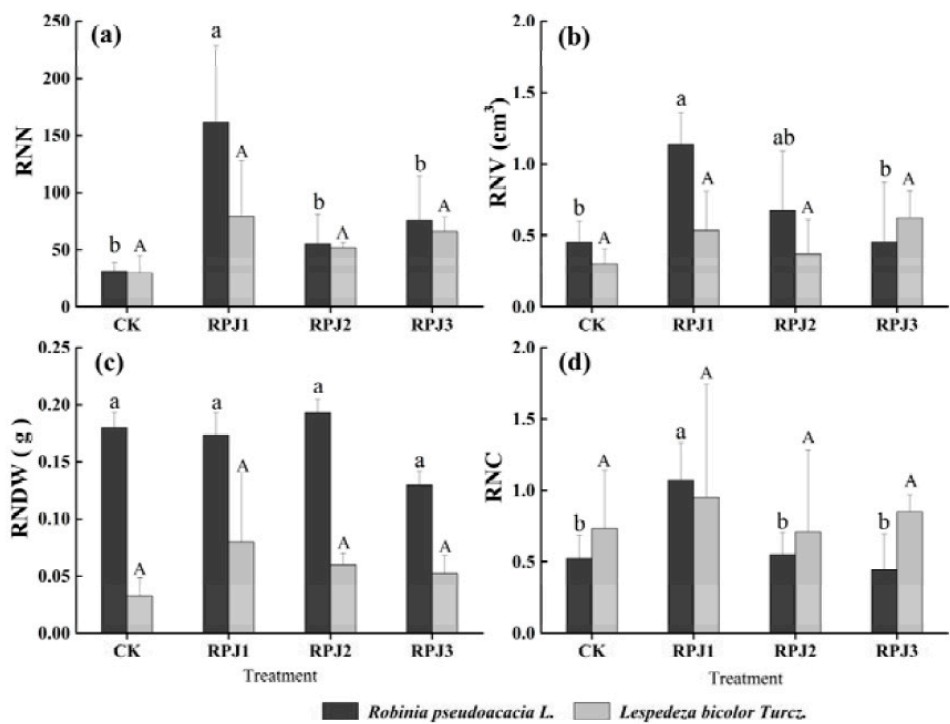

**Figure 2.** Changes in the root nodule number (**a**), root nodule volume (**b**), dry weight root nodules (**c**), and root nodule contributions (**d**) under the addition of efficient rock-solubilizing microbes. RNN, root nodules number; RNV, root nodules volume; RNDW, root nodules dry weight; RV, root volume; RNC, root nodules contribution. The data are presented as the mean $\pm$ SD ($n = 3$). One-way ANOVA (analysis of variance) was performed for each parameter. Different lowercase letters for *R. pseudoacacia* and capital letters for *L. bicolor* indicate a significant difference at $p < 0.05$ by Duncan's multi-range test.

Although the use of rock-solubilizing microbes had no significant effect on the nodulation of *L. bicolor*, it was still improved through the addition of microbial agents (Figure 2, $p > 0.05$).

### 3.3. Effects of Rock-Solubilizing Microbes on Soil Properties

Further to the quantification of the responses of plants to the addition of rock-solubilizing microbes, we investigated the influence of added microbial agents to soil properties (Figures 3 and 4). As can be seen, in contrast to no added rock-solubilizing microbes in *R. pseudoacacia* pots, the RPJ1 treatment significantly decreased the soil pH (Figure 3a, $p < 0.05$). The other two treatment groups had no significant effects on the soil pH (Figure 3a, $p > 0.05$). The RPJ2 treatment exhibited significantly increased total soil carbon in *R. pseudoacacia* pots (Figure 3b, $p < 0.05$). However, there were no significant impacts on the total soil carbon in *L. bicolor* pots following the use of rock-solubilizing microbes (Figure 3b, $p > 0.05$).

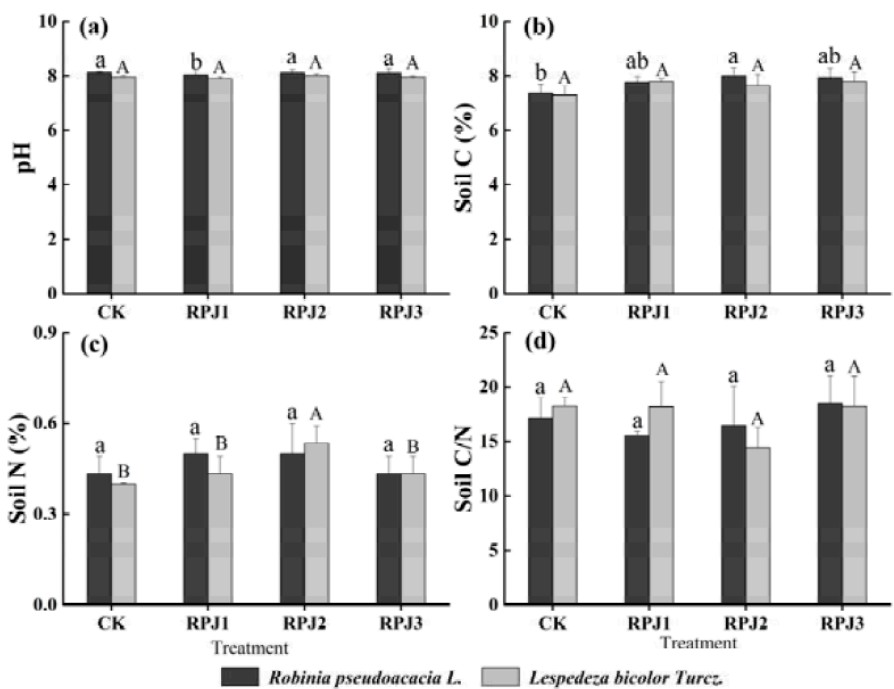

**Figure 3.** Changes in soil pH (**a**), total soil carbon (**b**), total soil nitrogen (**c**), and soil carbon to nitrogen ratio (**d**) under the addition of efficient rock-solubilizing microbes. The data are presented as the mean $\pm$ SD ($n = 3$). One-way ANOVA (analysis of variance) was performed for each parameter. Different lowercase letters for *R. pseudoacacia* and capital letters for *L. bicolor* indicate a significant difference at $p < 0.05$ by Duncan's multi-range test.

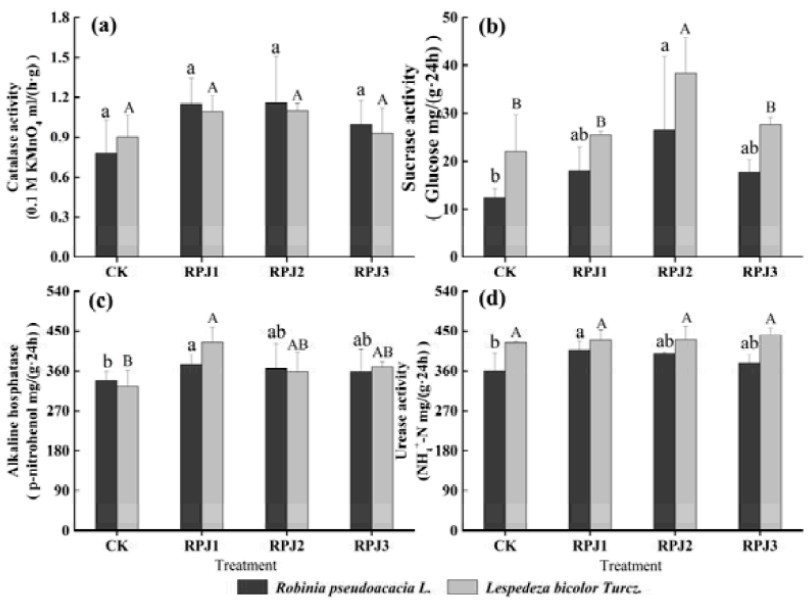

**Figure 4.** Changes in soil catalase activity (**a**), soil sucrose activity (**b**), soil alkaline phosphatase activity (**c**), and soil urease activity (**d**) under the addition of efficient rock-solubilizing microbes. The data are presented as the mean $\pm$ SD ($n = 3$). One-way ANOVA (analysis of variance) was performed for each parameter. Different lowercase letters for *R. pseudoacacia* and capital letters for *L. bicolor* indicate a significant difference at $p < 0.05$ by Duncan's multi-range test.

Added rock-solubilizing microbes had no significant effects on the total soil nitrogen in *R. pseudoacacia* pots (Figure 3c, $p > 0.05$). Nonetheless, the RPJ2 treatment significantly

improved the total soil nitrogen in *L. bicolor* pots (Figure 3c, *p* < 0.05). Additional rock-solubilizing microbes had no significant influence on the soil carbon to nitrogen ratio (Figure 3d, *p* > 0.05).

### 3.4. Effects of Rock-Solubilizing Microbes on Soil Enzymes

As shown in Figure 4a, the soil catalase activity in three of the microbial agent treatments and two types of plant were slightly increased; however, no significant differences were observed (*p* > 0.05). The RPJ2 treatment had significantly improved soil sucrase activity in *R. pseudoacacia* and *L. bicolor* pots (Figure 4b, *p* < 0.05). The other treatments marginally promoted the sucrase activity of the soil (*p* > 0.05).

The RPJ1 treatment significantly enhanced the activities of soil phosphatase and urease in *R. pseudoacacia* and *L. bicolor* pots (Figure 4c, *p* < 0.05). The other treatments had no significant effects on the activities of soil phosphatase (Figure 4c, *p* > 0.05). Further, the RPJ1 treatment exhibited significantly increased soil urease activity in *R. pseudoacacia* pots (Figure 4d, *p* < 0.05). However, added rock-solubilizing microbes had no significant influence on the soil urease activity in *L. bicolor* pots (Figure 4d, *p* > 0.05).

### 3.5. Boosted Regression Tree (BRT) Model Analysis

Among the plant nodule and soil characteristic indices, the greatest contributor to the height of *R. pseudoacacia* was soil sucrase activity at a rate value of 29.52%. The total root volume contributed 18.47% to the growth height of *R. pseudoacacia*, whereas the contribution rate of dry weight root nodules to the growth height of *R. pseudoacacia* was 12.1%, whereas other factors contributed less than 10% (Figure 5a).

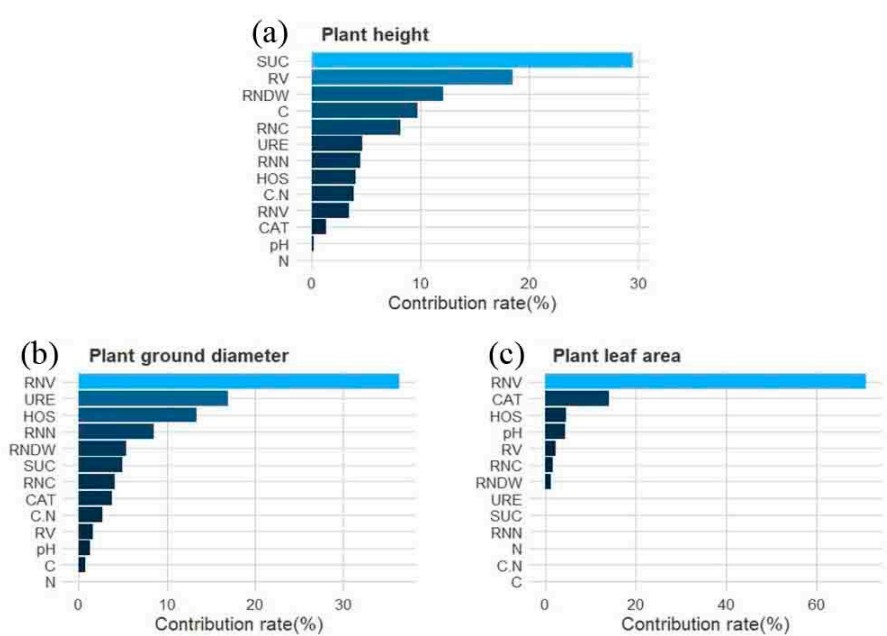

**Figure 5.** Boosted regression tree (BRT) model analyzed the contribution of soil factors and plant nodule conditions to the height (**a**), ground diameter (**b**) and leaf area (**c**) of *R. pseudoacacia*. RNN, root nodules number; RNV, root nodules volume; RNDW, root nodules dry weight; RV, root volume; RNC, root nodules contribution; CAT, soil catalase activity; SUC, soil sucrase activity; HOS, soil alkaline phosphatase activity; URE, soil urease activity; pH, soil pH; C, soil total carbon; N, soil total nitrogen; C/N, soil carbon to nitrogen ratio.

However, the root nodule volume contributed the most to the basal diameter and leaf area growth of *R. pseudoacacia*, at 35.04% and 71.08%, respectively. The contribution rate of soil urease activity to the basal diameter growth of *R. pseudoacacia* was 16.26%. (Figure 5b,c).

BRT analysis revealed that the root nodule number had the highest contribution rate (28.84%) to the growth height of *L. bicolor*, which was followed by total soil carbon and root volume, at 23.74% and 16.24%, respectively. Other factors contributed less than 10% to the tree height (Figure 6a). The contribution rate of total soil carbon to the basal diameter of *L. bicolor* was 33.82%, whereas those of soil sucrase activity and soil total nitrogen were 16.77% and 16.55%, respectively (Figure 6b).

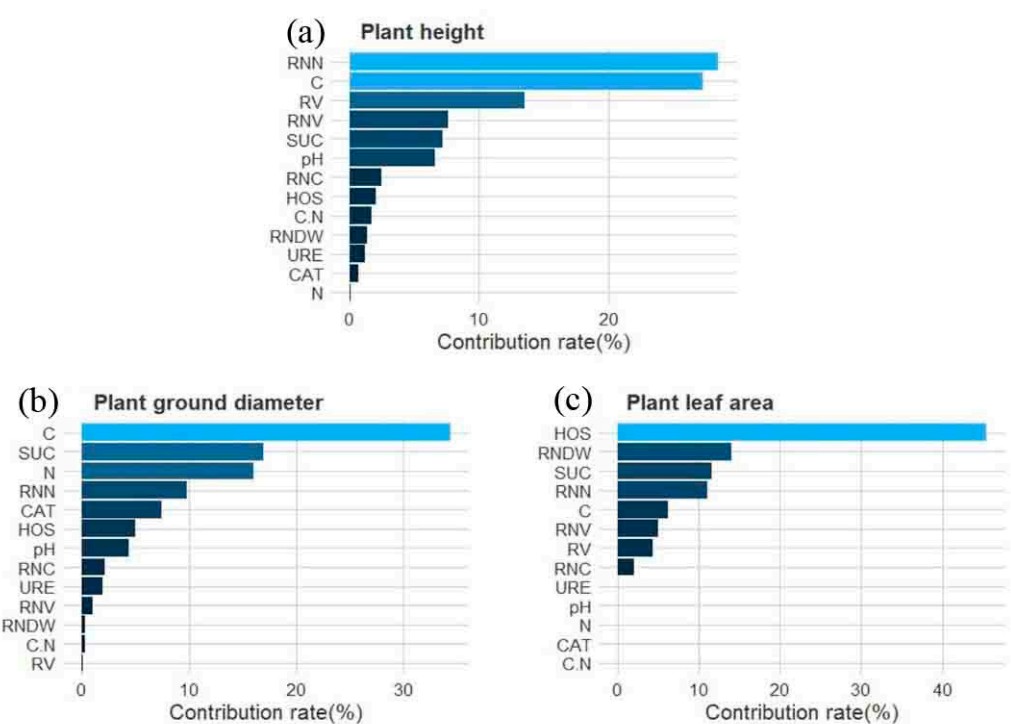

**Figure 6.** BRT model analyzed the contribution of soil factors and plant nodule conditions to the height (**a**), ground diameter (**b**) and leaf area (**c**) of *L. bicolor*. RNN, root nodules number; RNV, root nodules volume; RNDW, root nodules dry weight; RV, root volume; RNC, root nodules contribution; CAT, soil catalase activity; SUC, soil sucrase activity; HOS, soil alkaline phosphatase activity; URE, soil urease activity; pH, soil pH; C, soil total carbon; N, soil total nitrogen; C/N, soil carbon to nitrogen ratio.

Conversely, the contribution rate of soil phosphorus (indicated by soil phosphatase activity) to the leaf growth area of *L. bicolor* was higher than soil carbon and nitrogen, at a rate of 45.31%. The dry weight root nodule, soil sucrase activity, and root nodule number contribution rates to the leaf growth area of *L. bicolor* were 13.96%, 11.63%, and 11.02%, respectively (Figure 6c).

### 3.6. Correlation Coefficients between Plant Growth and Different Parameters

The height of *R. pseudoacacia* was significantly correlated to soil sucrase activity, whereas other soil enzyme activities had no significant impacts on the height of *R. pseudoacacia*. The basal diameter of *R. pseudoacacia* exhibited positive correlations primarily with the root nodule volume and root nodule number. The leaf area of *R. pseudoacacia* correlated positively with the root nodule number, suggesting that root nodules may play a critical role in the leaf growth of *R. pseudoacacia* (Figure 7a).

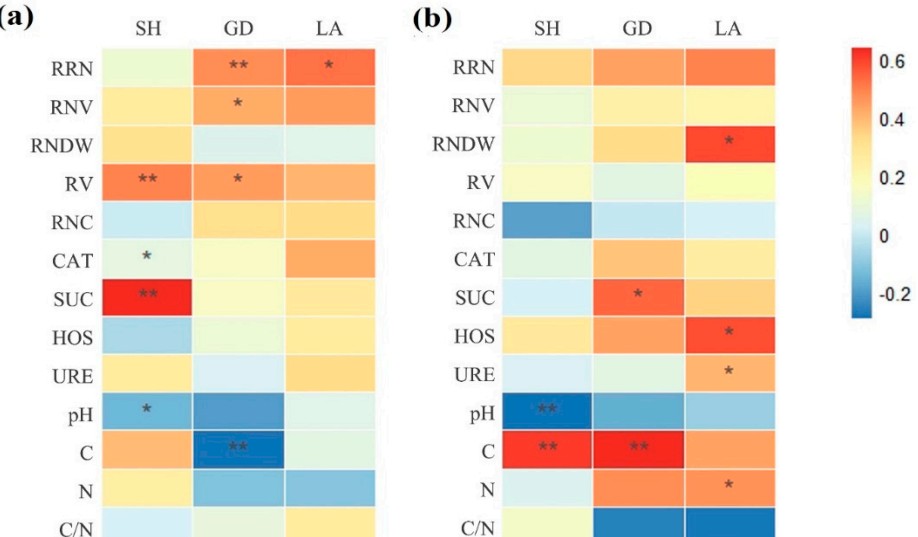

**Figure 7.** Pearson's correlation analyzed differences between the growth conditions of R. pseudo-acacia (**a**) and L. bicolor (**b**). Significant correlations are marked in red (positive) and blue (nega-tive). ** indicates significant correlation at $p < 0.01$; * indicates significant correlation at $p < 0.05$. SH, height; GD, basal diameter; LA, leaf areas; RNN, root nodules number; RNV, root nodules volume; RNDW, root nodules dry weight; RV, root volume; RNC, root nodules contribution; CAT, soil catalase activity; SUC, soil sucrase activity; HOS, soil alkaline phosphatase activity; URE, soil urease activity; pH, soil pH; C, soil total carbon; N, soil total nitrogen; C/N, soil car-bon to nitrogen ratio.

However, the height and basal diameter of *L. bicolor* primarily had positive correlations with the total soil carbon. The leaf areas of *L. bicolor* were positively correlated with the dry weight root nodules, soil phosphatase activity, soil sucrase activity, and total soil nitrogen (Figure 7b).

## 4. Discussion

In soil-plant systems, rhizospheric microbes interact with arbuscular mycorrhizal and ectomycorrhizal fungi, and play a critical role in determining the nutrition, growth, and health of plants [4,25–28]. Among the numerous interactions between plants and soil, microorganisms also play a key role [29,30], which supported our first hypothesis. Therefore, the application of efficient rock-solubilizing microbes not only promotes rock weathering but also has a certain impact on the growth of plants.

1. Effects of rock-solubilizing microbes on plant growth

Rock solubilizing microbial inoculums are typically based on harmless filtrated and identified microbial strains, which optimize soil microbial community structures on bare rock surfaces and accelerate the weathering of rock into soil without environmentally toxic effects. This is in contrast to compound fertilizers and binders that are commonly employed for mine restoration projects. The purpose of combining efficacious rock solubilizing microbes with soil spray-sowing substrates was to expedite rock weathering. However, our study revealed that the addition of microbial agents also had an effect on plant growth.

Plant height, basal diameter, and leaf area are important traits that influence the growth and sustainability of reestablished vegetation. It is also an important objective for ecological restoration and a critical indicator that represents the status of plant growth and nitrogen uptake at the vegetative stage. It is widely acknowledged that microbial interactions with plants improve their performance by enhancing mineral nutrition. This becomes possible due to the increased bioavailability of soil-originating nutrients. Plants depend on soil; however, plant-associated microorganisms also play a crucial role in the formation or modification of soil [31–33].

Boosted regression tree (BRT) analysis illustrated that the utilization of nitrogen and carbon contributed the most to the growth index of *R. pseudoacacia* in terms of tree height, basal diameter, and leaf area (Figure 5).

BRT analysis showed that plant nitrogen uptake, nodulation status, and soil urease activities contribute more than 10% to the growth factors of *R. pseudoacacia* and *L. bicolor*. The plant's utilization of carbon was primarily reflected through the height of *R. pseudoacacia* and the ground diameter of *L. bicolor*. It can be seen that uptake of nitrogen by plants has the greatest impact on their growth.

2. Effects of rock-solubilizing microbes on plant root nodules

The uptake capacity for nitrogen by plants directly influences the long-term effects of vegetation restoration. Studies have shown that nitrogen limits the long-term growth of plants [34], where nodulation significantly affects the efficiency of nitrogen fixation [35–37].

Studies have shown that inoculation with rhizobia can improve the nodulation of legumes roots and promote plant growth, as the most isolated bacteria in legumes nodules are not rhizobia [38]. These results signified that rhizobia can be a rich source of plant growth-promoting rhizobacteria (PGPR) [39]. In summary, the appropriate nodulation status of plants can promote plant growth.

For this study, we explored the relationships between plant growth and nodulation in pot experiments. Correlation analysis revealed that increased nodule populations can significantly promote the ground diameter and leaf areas of *R. pseudoacacia*, which are two important indices for plant growth. This corroborated that the growth of *R. pseudoacacia* was enhanced via the addition of rock-solubilizing microbial inoculums through the increased nodulation of *R. pseudoacacia*. Plant nodulation is facilitated by plant hormones [40], the rhizospheric environment [39,41], and plant-rhizobia signaling [42].

However, the growth of *L. bicolor* had a significant positive correlation with the carbon content of the soil and had no association with the nodulation status of *L. bicolor*. A potential rationale for this phenomenon was that carbon is the main growth-limiting factor for *L. bicolor*. Nevertheless, the growth of *L. bicolor* was improved following the addition of rock-solubilizing microbes, where the height and basal ground area of *L. bicolor* are significantly increased under the RPJ1 treatment.

3. Effects of rock-solubilizing microbes on soil C and N content

Soil urease activity characterizes the status of soil nitrogen [43,44]. Increased rhizospheric soil urease activities are positive for plants in terms of their utilization of soil nitrogen [45].

The results revealed that the carbon and nitrogen supply capacities of the soil had a significant influence on the basal diameter growth of *L. bicolor*. Studies have shown that the utilization of plant nitrogen is related to plant nodulation. Addition of defined microorganisms can strongly affect the physicochemical properties of soils and consequently plants. Therefore, our study revealed that the addition of rock-solubilizing microbial inoculums has a positive impact on plant nitrogen utilization efficiency.

For future work, it is suggested that the responses of soil microbial community structures to the application of rock-solubilizing microbial inoculums should be investigated. Furthermore, we want to explore the kinetics behind the changes in plant root nodulation under different microbial treatments. Leguminous plants with shallow-roots and potent soil-fixing capacities have been extensively employed for the remediation of damaged ecosystems due to mining [46]. Our study will not only put forward a new idea of the long-term maintenance of restored vegetation and ecosystems at former mining sites, but also provide improved guidance for the application of microbial agents under different site conditions in the future.

Under site conditions where it is difficult for vegetation to survive, leguminous plants exhibit a stronger resistance to stress. The main reason may be that they can symbiotically co-exist with rhizobia in the soil to enhance their own nitrogen fixation capacities, particularly toward alleviating site conditions such as bare rocky slopes that

lack nitrogen. Therefore, we aim to continue to explore the responses of legume nodulation to the application of rock-solubilizing microbial inoculums in future studies.

## 5. Conclusions

Our study suggests that the application of rock-solubilizing microbial inoculums is an effective technology for the restoration and sustainability of vegetation at damaged carbonate mine sites. It was observed that the applied rock-solubilizing microbial inoculums promoted the growth of *R. pseudoacacia* and *L. bicolor*, which are plants that are commonly used for the restoration of mine site vegetation, while expediting the weathering of rocks. Rock-solubilizing microbial inoculums can enhance the nodulation of plants, while improving the rhizospheric soil environment to promote plant growth. In particular, when nitrogen is the main growth-limiting factor, the impact of rock-solubilizing microbial inoculums on plant growth is more obvious. Further investigations should be done using molecular biology technologies, to explore the response mechanisms of plant growth under the application of rock-solubilizing microbial inoculums.

**Author Contributions:** J.Z. and Z.J. conceived the experiments. Z.J., M.M., C.L., X.L., S.M. and X.C. conducted the experiments. Z.J. and L.Z. interpreted data. Z.J. wrote the manuscript. B.Z. and L.Z. revised the manuscript. All authors have read and agreed to the published version of the manuscript.

**Funding:** This project was supported by the Jiangsu Agricultural Science and Technology Innovation Fund (Grant No. CX(17)1004), National Special Fund for Forestry Scientific Research in the Public Interest (Grant No. 201504406), the Priority Academic Program Development of Jiangsu Higher Education Institutions (PAPD), and Financial support for this study by the China Postdoctoral Science Foundation (2018M642260).

**Institutional Review Board Statement:** Not applicable.

**Informed Consent Statement:** Not applicable.

**Data Availability Statement:** The data presented in this study are available on request from the corresponding author.

**Acknowledgments:** We are grateful to Frank Boehm, who works at Lakehead University in Canada, for the editing of this paper and correcting the English.

**Conflicts of Interest:** The authors declare that they have no known competing financial interests or personal relationships that could have appeared to influence the work reported in this paper.

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
