# Peer review of "Rock-Solubilizing Microbial Inoculums Have Enormous Potential as Ecological Remediation Agents to Promote Plant Growth"

_forests, doi:10.3390/f12030357_

Round 1

Reviewer 1 Report

Dear authors,

It is really interesting work with significant potential.

I recommend to make some corrections as follows:

1) Read the Instruction for authors - In the text all references should be mentioned as number in brackets [1], etc. So in the Reference section too.

2) The description of Figs 1, 2, 3 and 4 -  Latin names of species should be in italic

3) Line 100-109 - The methodology for microbial agents is not clearly described and no reference is cited.

4) Line 111-114 - It is not clear why you have mixed such a media. Please, explain and give some references

5) Line 138-140 - There are no references for pH measurent and analyses of C and N in soil 

Author Response

Dear reviewers,

Thank you very much for your letter and the reviewer’s comments concerning our manuscript entitled “Rock solubilizing microbial inoculums have enormous potential as ecological remediation agents to promote plant growth” (ID: forests-1123357). Those comments are all valuable and very helpful for revising and improving our paper. We have carefully revised the manuscript according to each comment referred by the reviewers. The revised portions were highlighted in blue in the marked-revised manuscript.

Reviewer #1:

Point 1: Read the Instruction for authors - In the text all references should be mentioned as number in brackets [1], etc. So in the Reference section too.

Response: Thanks for your suggestions. We have revised the format of references and citations according to the instructions for authors. Please see the changed marked manuscript.

Point 2: The description of Figs 1, 2, 3 and 4 -  Latin names of species should be in italic.

Response: Thank you for the useful comments. We have change We changed the Latin names of species to italics in the description of figure. Please see lines 179, 196, 216, 235-236, 249, 271 and 287-288 in manuscript with mark.

Point 3: Line 100-109 - The methodology for microbial agents is not clearly described and no reference is cited.

Response: Thanks for your positive comments. We have redescribed the process of making microbial agents and added references. Please see lines 101-105 in manuscript with mark.

Point 4: Line 111-114 - It is not clear why you have mixed such a media. Please, explain and give some references.

Response: Thanks for your positive comments. Our pot experiment is a combination of simulating microbial technology and soil spray-sowing technology, so the substrate in our pot experiment is arranged according to the proportion of the spray-sowing substrate configuration. Please see lines 111-116 in manuscript with mark.

Point 5: Line 138-140 - There are no references for pH measurement and analyses of C and N in soil.

Response: Thanks for your positive comments. We have added references for pH measurement and analyses of C and N in soil. Please see lines 144 in manuscript with mark.

We have tried our best to improve the manuscript and have changed some descriptions. These descriptions will not alter the content and framework of the paper. We are very appreciated for Editors/Reviewers’ work, and we sincerely hope that these revisions will meet requirements. Once again, thank you very much for your work and comments.

Yours sincerely,

Jinchi Zhang

Reviewer 2 Report

Dear Authors,

Your study about the impact of different microbial inoculums on plant growth with respect to ecological soil remediation in mining areas is interesting and should be shared with the community. But in my opinion some revision is needed before publication.

General Remarks

Your manuscript generally is well organized and has a reasonable length. The use of English language generally is good, however at places in my opinion needs some revision.

Further I had some problems with the experimental setup. Maybe I am wrong, but I missed a reference sample where your test plants were grown in the same soil material under the same conditions, but without addition of an inoculum. The Results section compares plant factors (growth, height, leaf area, etc.), however not with reference to plants grown in non-treated soil. As to the soil used itself, here I miss some more information, e.g. about soil type and texture. Especially soil texture impacts plant growth at it determines important aspects like water availability. Further, I would think a table listing all tested parameters (pH, C, N, enzymes) of the control (i.e. soil without added inoculum) and of the different variants (i.e. the different inoculums added) would be helpful and would improve the manuscript.

Please check the reference style of the journal, I would think numbers should be given instead of names in the text!

Further, please check the font size used in your figures, they mostly are too small!

Overall, given the number of points that in my opinion need to be considered, I would recommend some major revisions before publication.

Special Remarks

For my Special Remarks, please see the attached pdf document I added my comments to!

Author Response

Thank you very much for your letter and the reviewer’s comments concerning our manuscript entitled “Rock solubilizing microbial inoculums have enormous potential as ecological remediation agents to promote plant growth” (ID: forests-1123357). Those comments are all valuable and very helpful for revising and improving our paper. We have carefully revised the manuscript according to each comment referred by the reviewers. The revised portions were highlighted in blue in the marked-revised manuscript.

General Remarks

Point 1: Your manuscript generally is well organized and has a reasonable length. The use of English language generally is good, however at places in my opinion needs some revision.

Response: Thank you for the positive comments. We will will revise each item carefully according to your suggestion.

Point 2: Further I had some problems with the experimental setup. Maybe I am wrong, but I missed a reference sample where your test plants were grown in the same soil material under the same conditions, but without addition of an inoculum.

Response: Thank you for the useful comments. The description of our experimental design is not rigorous enough. We used fermented microbial agents in our experimental group, and sterilized microbial liquid culture medium was used in the control group. Because this could eliminate the effects of nutrient elements in liquid culture medium on soil and plants. We modified this part. Please see lines 121-123 in manuscript with mark. Thank you again for your suggestion.

Point 3: The Results section compares plant factors (growth, height, leaf area, etc.), however not with reference to plants grown in non-treated soil.

Response: Thanks for your comments. In the result part, CK group represents the reference group which soil without microbial addition. And we added sterilized microbial liquid culture medium to highlight the role of functional microorganisms rather than nutrients in microbial liquid culture medium. Thank you again for giving us a chance to explain this problem to you.

Point 4: As to the soil used itself, here I miss some more information, e.g. about soil type and texture. Especially soil texture impacts plant growth at it determines important aspects like water availability.

Response: Thank you for the positive comments. We have added soil texture and type as you suggested Please see lines 112-114 in manuscript with mark. Thank you again for your suggestion.

Point 5: Further, I would think a table listing all tested parameters (pH, C, N, enzymes) of the control (i.e. soil without added inoculum) and of the different variants (i.e. the different inoculums added) would be helpful and would improve the manuscript.

Response: Thanks for the useful comments. We would like to add a table according to your request. However, we found in the process of making changes that the images showed the difference more intuitively. So, we kept the images and adjusted the fonts to a suitable size for reading.

Point 6: Please check the reference style of the journal, I would think numbers should be given instead of names in the text!

Response: Thanks for your suggestions. We have revised the format of references and citations according to the instructions for authors. Please see the changed marked manuscript.

Point 7: Further, please check the font size used in your figures, they mostly are too small!

Response: Thanks for the useful comments. We have adjusted the font size in the figure for the convenience of readers. Thank you again for giving us a chance to modify this problem.

Special Remarks

Point 8: For my Special Remarks, please see the attached pdf document I added my comments to!

Response: Thank you for your attachment. In this There are many excellent suggestions in the pdf document that can be used to improve the manuscript. We have revised the manuscript one by one according to the attachment. Thank you again for suggestion.

We have tried our best to improve the manuscript and have changed some descriptions. These descriptions will not alter the content and framework of the paper. We are very appreciated for Editors/Reviewers’ work, and we sincerely hope that these revisions will meet requirements. Once again, thank you very much for your work and comments.

Yours sincerely,

Jinchi Zhang

Round 2

Reviewer 2 Report

Dear Authors,

You worked very much on your manuscript which improved considerably! You did a great job!

However, I still found some points I think need to be considered. This would mean some more minor revisions before publication. In this context a special focus should be on defining the reference sample which I still somehow miss in the text! Or did I miss something? Further, at one place (Line 136) still “biochar” is mentioned, however, biochar is not mentioned in the description of the experimental setup!

Special Remarks

For my Specific Remarks, please see the attached pdf document I added my comments to!

Author Response

Thank you very much for your letter and the reviewer’s comments concerning our manuscript entitled “Rock solubilizing microbial inoculums have enormous potential as ecological remediation agents to promote plant growth” (ID: forests-1123357). Those comments are all valuable and very helpful for revising and improving our paper. We have carefully revised the manuscript according to each comment referred by the reviewers. The revised portions were highlighted in blue in the marked-revised manuscript.

Reviewer #2 (Round 2):

Point 1: This would mean some more minor revisions before publication. In this context a special focus should be on defining the reference sample which I still somehow miss in the text! Or did I miss something?

Response: Thanks for your suggestions what will optimize this manuscript greatly. We carefully checked for the description of experimental design. We confirmed that the reference sample you want us to define has been described in detail on line 118. Please see the manuscript with marked. Perhaps I didn't catch your meaning. If the reply doesn't explain your doubt, please don't hesitate to point it out.

Point 2: Further, at one place (Line 136) still “biochar” is mentioned, however, biochar is not mentioned in the description of the experimental setup!

Response: Thank you for the useful comments. This place has been corrected accordingly. This is a big error when occurred for polishing our manuscript. Thank you very much for pointing it out.

Point 3: For my Specific Remarks, please see the attached pdf document I added my comments to!

Response: Thank you for your attachment. In this There are many excellent suggestions in the pdf document that can be used to improve the manuscript. We have revised the manuscript one by one according to the attachment. Thank you again for suggestion.

We have tried our best to improve the manuscript and have changed some descriptions. These descriptions will not alter the content and framework of the paper. We are very appreciated for Editors/Reviewers’ work, and we sincerely hope that these revisions will meet requirements. Once again, thank you very much for your work and comments.

Yours sincerely,

Jinchi Zhang
